# Access to public sector family planning services and modern contraceptive methods in South Africa: A qualitative evaluation from community and health care provider perspectives

Yolandie Kriel[1,2]*, Cecilia Milford[1], Joanna Paula Cordero[4], Fatima Suleman[3], Petrus S. Steyn[4], Jennifer Ann Smit[1]

1 MRU (MatCH Research Unit), Department of Obstetrics and Gynaecology, Faculty of Health Sciences, University of the Witwatersrand, Durban, South Africa, 2 School of Public Health and Nursing, College of Health Sciences, University of KwaZulu-Natal, Durban, South Africa, 3 Discipline of Pharmaceutical Science, College of Health Sciences, University of KwaZulu-Natal, Westville, South Africa, 4 Department of Sexual and Reproductive Health and Research, UNDP-UNFPA-UNICEF-WHO-World Bank Special Programme of Research, Development and Research Training in Human Reproduction (HRP), World Health Organization, Geneva, Switzerland

* ykriel@mru.ac.za

**Data Availability Statement:** The database is not publicly available as it contains information that could compromise research participants' privacy

## Abstract

Progress has been made to improve access to family planning services and contraceptive methods, yet many women still struggle to access contraception, increasing their risk for unintended pregnancy. This is also true for South Africa, where over fifty per cent of pregnancies are reported as unintended, even though contraception is freely available. There is also stagnation in the fertility rate indicators and contraceptive use data, indicating that there may be challenges to accessing contraception. This paper explores the evaluation of access to contraception from community and health care provider perspectives. This qualitative study explored factors affecting the uptake and use of contraception through focus group discussions (n = 14), in-depth interviews (n = 8), and drawings. Participants included male and female community members (n = 103) between 15 and 49 years of age, health care providers (n = 16), and key stakeholder informants (n = 8), with a total number of 127 participants. Thematic content analysis was used to explore the data using NVivo 10. Emergent themes were elucidated and thematically categorised. The results were categorised according to a priori access components. Overall, the results showed that the greatest obstacle to accessing contraception was the accommodation component. This included the effects of integrated care, long waiting times, and limited operational hours–all of which contributed to the discontinuation of contraception. Community members reported being satisfied with the accessibility and affordability components but less satisfied with the availability of trained providers and a variety of contraceptive methods. The accessibility and affordability themes also revealed the important role that individual agency and choice in service provider plays in accessing contraception. Data from the illustrations showed that adolescent males experienced the most geographic barriers. This study illustrated the importance of examining

and consent. However, some anonymised aspects of the datasets may be available upon request and with the permission of the Department of Sexual and Reproductive Health and Research, World Health Organization, and the MatCH Research Unit (MRU). Note that data sharing is subject to WHO data sharing policies and data use agreements with the participating research centre, MRU. For permission to access the database, please contact the MatCH Research Unit at info@mru.ac.za.

**Funding:** This work received funding through a grant received by UNDP-UNFPA-UNICEF-WHO-World Bank Special Programme of Research, Development and Research Training in Human Reproduction (HRP), a cosponsored programme executed by the World Health Organization (WHO) [Award 53405] from the Bill and Melinda Gates Foundation [OPP1084560] and the United States Agency for International Development (USAID) through the USAID/WHO Umbrella Grant 2016-2018. The funders had no role in the study design, data collection and analysis, decision to publish, or preparation of the manuscript.

**Competing interests:** The authors have declared that no competing interests exists.

**Abbreviations:** CYPR, Couple's year of protection rate; CPR, Contraceptive prevalence rate; DMPA, Depomedroxyprogesterone; FP, Family Planning; HCP, Health care provider; KI, Key Informant; KZN, KwaZulu-Natal; LMICs, Low- and middle-income countries; mCPR, Modern contraceptive prevalence rate; SRH, Sexual and Reproductive Health; SDP, Service Delivery Point; SA, South Africa; TFR, Total fertility rate.

access as a holistic concept and to assess each component's influence on contraceptive uptake and use.

## Introduction

Family planning (FP) services and modern contraceptive methods are some of the most important public health strategies that can improve the lives of individuals, families, communities, and nations. Access to sexual and reproductive health (SRH) services, including family planning (FP) programmes and modern contraception, and contraception is a human right and a target set out by the Sustainable Development Goals (SDG) target 3.7 [1, 2]. At the London Summit in 2012, Family Planning 2020 (FP2020) was established with the goal to get an additional 120 million users onto contraception [3]. The SDG target, FP2020, along with General Comments No. 14 and No. 22 set out by the United Nations Committee on Economic, Social, and Cultural Rights [4], are driving goals for the international family planning community and nations, including South Africa (SA), to ensure equitable access to SRH services and contraceptive methods.

Although much progress has been made over the past couple of decades many women in low-and-middle-income countries (LMICs), including women in SA, struggle to access SRH services and contraception [3, 5, 6]. An estimated 230 million women and adolescent girls need modern contraception [3, 7] in LMICs, while Sub-Saharan Africa remains the region with the highest fertility rate and unmet need for contraception [8–10].

Compared with other countries in the Sub-Saharan region, SA has an advanced total fertility rate (TFR, 2.6), relatively low unmet need for married women (14.9%), moderate modern contraceptive prevalence rate (mCPR, 47.9%), and high levels of general knowledge about available contraceptive methods amongst men and women [11, 12]. However, numerous SRH challenges exist. Amongst these are high levels of unintended pregnancies (over half of all pregnancies are unintended), a high rate of unmet need amongst unmarried young women (27.9%), adolescent pregnancies (16% of women aged 15–19 years have had a pregnancy), high incidence of termination of pregnancies, and an alarming increase in the pregnancy-related mortality ratio from 150 deaths per 100 000 live births in 1998 to 536 deaths per 100 000 live births in 2016 [5, 6, 12].

There is also a noticeable stagnation in contraceptive use and fertility trend rates in SA over the past two decades. SA's total fertility rate has declined from an estimated 6.0 in 1960 to 2.9 in 1998 [12–14]. But between 1998 and 2016, the TFR has remained almost stagnant, dropping to 2.6 in 2016. Except for the total demand for contraception, met need, and unmet need, which all increased, the stagnating trend can be seen across all the other fertility and contraceptive use indicators (Fig 1) [12, 14].

These stagnating trends point to challenges accessing SRH services and contraception, despite SA having one of the longest-running modern family planning programmes in Sub-Saharan Africa [11]. Providing SRH services and contraception is also supported by progressive policies and guidelines [5]. Modern contraception is provided free of charge through the public health sector at primary health care (PHC) clinics that operate from 08:00 am to 4:00 pm on weekdays.

The South African District Health Survey (SADHS, 2016) data report that 2% of contraceptive users discontinued their method due to access-related barriers, including affordability and physical access [12]. Only focusing on affordability and physical access contributes to an

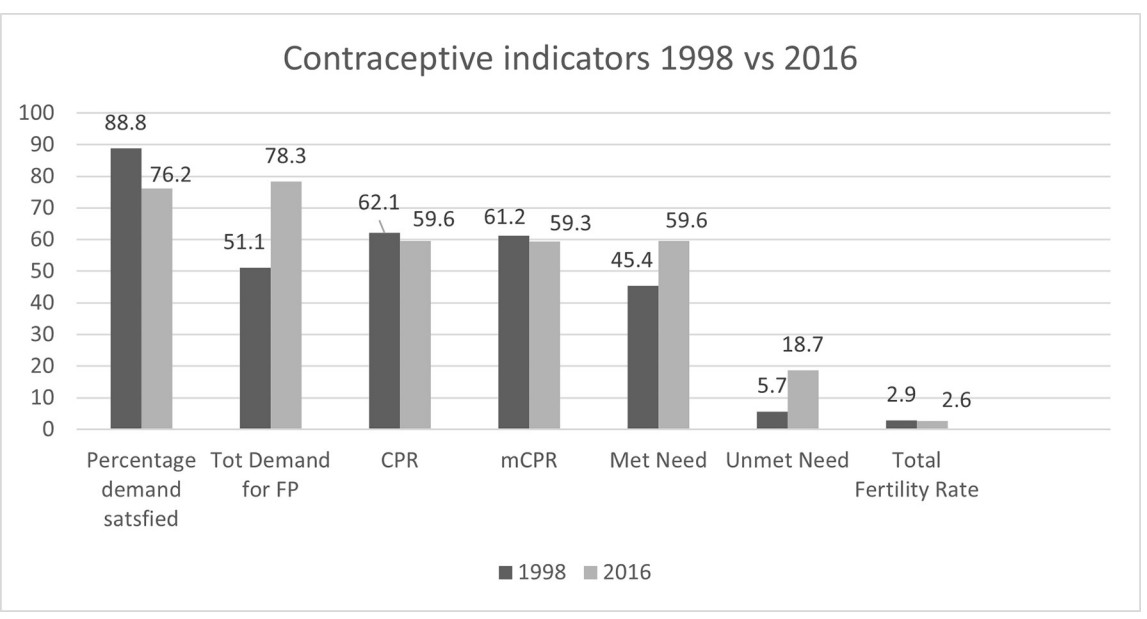

**Fig 1. Contraceptive indicators between 1998 and 2016 (SADHS, 2016).**

inaccurate understanding of access. Other sources have identified additional barriers, including poor quality of care (QoC), limited variety of modern contraceptive methods, fragmented integration of sexual and reproductive health (SRH) services with PHC services, ailing infrastructure, shortage of trained nurses, poor attitudes of nurses, and stigma [5, 15–19].

Furthermore, Hasumi and Jacobsen [20] reported that 43.8% of South Africans who relied on the public health care sector experienced problems accessing care–which is concerning since an estimated 80% of the population rely on the public sector for the health care needs [20, 21]. High inequality levels to accessing health have also been reported, where aspects such as race, economic status, and gender influences access to care [21, 22].

Most women (nearly 83%) obtain their contraceptive method from the public healthcare sector, compared to only 11.4% who access contraception through private health care[5, 6, 12]. Except for male condoms and female sterilisation, most of the methods; 3-month injectable (94%), 2-month injectable (95%), implants (94%) and pills (77%); are accessed from government primary health care clinics or community health centres [12]. Fifty-one percent of male condoms are obtained from primary health care clinics/community health care centres, 26% from shops, and 12% from pharmacies. Female sterilisation is predominantly accessed through government (63%) and private (27%) hospitals [12]. The source of contraception plays an important part in which contraceptive methods are used, as not all methods are available at all facilities and rely on the qualification and specialisation of healthcare professionals [23].

On a national policy level, the SA government is dedicated to improving access to SRH services and modern contraception by committing to the goals set out by FP2030 [24]. The SA government has restructured contraceptive services from a specialised vertical programme to a PHC district-based integrated programme [25]. Progressive SRH policies were implemented to address inadequacies in contraceptive access due to past political history [5, 26, 27]. However, how effective these policies are at improving access to contraception is unclear. Integrating services—especially SRH services with HIV care is a crucial strategy proposed to improve access, inequity, and QoC [17, 28]. This area has received much attention in recent years, and there is a considerable drive to ensure that integration occurs [3, 16, 29, 30]. However, barriers

to accessing health care services (as mentioned above) must be addressed for integration to be effective [31].

Despite the enabling policies and restructuring, the SA public health sector has struggled to make real gains in improving access to services, including FP service [20, 21, 32]. Where these challenges lie within the larger concept of access and which components are most significant remains unclear.

Qualitative, exploratory data can provide critical understandings about factors influencing access [33], which can lead to the development of community-focused interventions to improve uptake and met need for modern contraception. This paper aims to explore each access component and the factors that influence the uptake and use of contraception from a community and health care provider (HCP) perspective.

## Methodology

### Theoretical framework

In this study, we used Penchansky and Thomas's [34] Access framework, which defines access as "the general concept which summarises a set of more specific areas of fit between the patient and the health care system" (p.128). Five interrelated components of access are used in this framework, namely accessibility, availability, affordability, accommodation, and acceptability [34].

*Accessibility* refers to the "relationship between location of supply and the location of clients"; *availability* refers to "the relationship of the volume and type of existing services (and resources) to the clients' volume and types of needs"; *accommodation* refers to the "relationship between the manner in which the supply resources are organized to accept clients [. . .] and the clients' ability to accommodate to these factors and the clients' perception of their appropriateness"; *affordability* refers to the cost of services related to the clients' ability to pay; and *acceptability* which refers to the relationship of the attitude between clients' and providers [34: 128].

Versions of these components are found in later frameworks, such as the UNCESCR (United Nations Committee on Economic, Social, and Cultural Rights) availability, accessibility, acceptability, and good quality (AAAQ) framework [35, 36], or the FP focused access frameworks proposed by Choi et al [33], and Bertrand et al [37]. Yet, when employed, the original access framework continues to provide the most comprehensive exploration of access [5, 38–40]. Furthermore, within the SRH literature, it is important to distinguish between access and QoC. Ambiguity and overlap exist between these two concepts within the literature [33, 37, 41], resulting in skewed understandings of either concept.

Social constructionism was used as the overall lens to view the data. As a broad theory, social constructionism postulates that humans construct concepts through which they can understand reality [42, 43]. These constructs are often deeply embedded within respective cultures and shape the way humans understand and make sense of the world they live in [43]. Adopting a social constructionist stance towards the study of FP services and contraceptive use allows for a variety of voices to be heard. It is beneficial when the opinions and knowledge of community members are sought as it allows for the expression of ideas and understanding by a variety of people.

### Setting

The study reported in this paper was conducted in the eThekwini District of KwaZulu-Natal (KZN) province, SA. KZN is the second-most populous province in SA and is situated on the country's east coast. The couple-years of protection rate (CYPR), a proxy indicator for the

CPR, for KZN has consistently fallen below the national median, with the 2018/19 CYPR being 44.4%—a significant drop from the 2016/2017 CYPR of 66.1% [44]. KZN also has the largest number of people living with HIV in the country, where the HIV prevalence rate is 27%, with over two million of the seven and a half million people living with HIV residing in the province [45, 46]. The eThekwini District is the third-largest metropolitan district in SA, with an estimated population size of three and a half million people. Similar to the KZN province, eThekwini also has the largest number of people living with HIV per district in the country [45].

## Study design

This study was conducted as part of formative work to inform the development of an intervention that aimed at increasing met need for modern contraception through community and HCP participation from a human rights perspective in SA, Kenya and Zambia (the UPTAKE Project). Overall study results are presented by Cordero et al. [47]. The detailed teamwork qualitative methodology is published elsewhere [48]. In this article, we report on findings from the South African data.

## Data collection

We conducted in-depth interviews (IDIs, n = 8) and focus group discussions (FGDs, n = 14) between 2015 and 2016 with a total of 127 participants. Purposive snowball sampling was used as the recruitment strategy. Fourteen FGDs were conducted. Twelve of these FGDs were with community participants (n = 103), and two with HCPs (n = 16). Six of the female FGDs were equally split into groups of adults (35 to 49 years), young adults (20 to 34 years), and adolescents (15 to 19 years) from rural (n = 27) and urban (n = 25) settings. The male FGD participants were also split into groups of adults (n = 7), young adults (n = 8), and adolescents (n = 10) but were mixed from both the rural and urban areas. The remaining three FGDs consisted of unmarried females (20 to 34 years n = 8), married/in-union females (20 to 34 years n = 10), and females without children (18 to 49 years n = 8), respectively. The HCP FGDs consisted of Group 1 (management and professional nurses, n = 8) and Group 2 (enrolled nurses, counsellors, and other operational staff, n = 8). The HCPs were recruited from clinics based in the two areas where the community participants resided.

IDIs were conducted with eight key informants (KIs) who were purposively selected based on their expertise in SRH services or community involvement. Four KIs were based within the communities from which the community members were recruited and consisted of two community caregivers, one educational specialist, and one traditional healer. The other four KIs held senior or programme level positions, specialising in SRH and FP.

FGDs and IDIs were conducted by trained research team members who had experience with qualitative interviewing and were fluent in the local languages (isiZulu and English). Interviewers were appropriately matched with gender, age, and professional position to facilitate discussions. Demographic data were collected from all participants and were descriptively analysed. FGDs were conducted at community-based facilities. Key informants were interviewed at locations convenient to them, ranging from research site offices to participants' homes.

## Ethics approval and consent to participate

This study received World Health Organization (WHO) Ethics Review Committee (ERC) (Project ID A65896) and Research Project Review Panel (RP2) approval. The University of the Witwatersrand Human Research Ethics Committee (Health- HREC, reference number

M1504101) provided local country ethics review and approval. The University of KwaZulu-Natal's Biomedical Research Ethics Committee (BREC) provided further reciprocity. The KwaZulu-Natal Provincial Department of Health granted permission for health care providers to be interviewed. All participants voluntarily signed an informed consent form, including permission to audio record the interview/group sessions. Identified parents or legal guardians provided consent for minors (those aged <18 years). The minors provided assent to participate in the study.

## Data collection and analysis

During the FGDs and IDIs, participants were asked to explore their views about the uptake of modern contraception. Interview guides contained key theme-specific questions that were tailored for each category or type of participant, including the female FGDs, male FGDs, HCP FGDs, and key informant IDIs (see S1 Data to S4 Data to review the interview guides and questions used in this study). Similar key thematic questions explored understandings of family planning; knowledge, attitudes, and practices; key barriers and enablers to family planning access; perceptions and definitions of QoC; and the role of community participation in family planning and contraceptive services. The collaborative nature of FGDs allowed for rich data to be collected and for shared and diverse opinions to be explored [49]. IDIs were used to interview key informants who had expert knowledge about the delivery of contraception, such as policy makers and senior health care professionals. Due to the lengthy nature of FGDs, IDIs were more suited to these types of participants.

The FGD participants were also asked to make simple drawings to capture their experiences of accessing or providing contraception. Key elements asked to be included were their primary source of obtaining contraception, the time travelled to the service, and their method of transport. They could also include any other aspects that they felt were relevant. The drawings allowed for an individual perspective to be explored in the data set during the group activity. Participants captured their own experiences and constructs about accessing contraception in their community.

Audio recordings of the FGDs and IDIs were transcribed and translated from isiZulu to English where necessary. The transcripts were reviewed and checked for accuracy and any ambiguity in the translations were discussed and clarified to ensure accuracy of the translations.

The audio transcripts and drawings underwent the same process of analysis. Firstly, a subset of transcripts and drawings were read, reviewed, and tested before a cross-country team of researchers developed a master codebook. During the initial coding round, a subset of coded transcripts was double coded to increase the validity, reliability, and consistency of the findings [48]. Additional coding was done on the individual country level to elucidate themes specific to each country. NVivo (version 10, QSR International) was used as the computer-assisted qualitative data analysis software that facilitated coding and thematic analysis of the audio transcripts and drawings. A detailed account of the teamwork methodology is presented by Milford et al. [48].

Thematic content analysis was used to identify *a priori* and emergent themes. *A priori* themes were based on components from Penchansky and Thomas's (1981) access framework described above. Inductive and emergent themes were further explored using the constant comparison method to allow for themes to emerge once data saturation was reached [50]. Coding queries and matrix coding queries were run to explore the data further. Matrix coding queries were used to elucidate comparative themes between the various groups of participants and to show where coding queries intersected [51]. These more complex queries allowed for

in-depth exploration and to uncover larger themes embedded in the data. The drawings were also analysed using the same codebook and access framework codes. Using the same codes allowed for data triangulation between what participants reported and what they captured in their drawings.

At the end of the project, the results were shared with community members, and there was a high degree of agreement with the results presented. This further contributed to the validity and accuracy of the data.

## Results

### Demographic results

A total of 103 community members participated in this study. Seventy-eight community participants were female (75%), and 25 were male (25%). The mean age of female participants was 26.4 years and males 23.8 years. Over half (53%) were in a long-term relationship but not living with their partner. At the time of data collection, 83.5% of the participants had used a modern contraceptive method.

Reported adolescent pregnancies were high, with seven (37%) out of the nineteen adolescent females reporting pregnancy. Five out of the nine adolescent girls in the urban adolescent group reported pregnancy, compared to two out of ten in the rural adolescent group. No male adolescents reported a pregnancy with a female partner. Table 1 below shows the demographic details for the community participants.

HCP FGD group 1 (management and professional nurses) consisted of six female and two male participants. All the participants were females in HCP FGD group 2 which consisted of enrolled nurses, counsellors, and other operational staff (n = 8). The Key Informants consisted of seven females and one male. All HCPs interviewed in this study worked at public PHCs or hospitals. The average years of experience were ten.

### Thematic results

This section explores the main themes that emerged from the data concerning questions about access. These emergent themes were categorised per component of access. Themes included: the current and preferred sources of FP services and modern contraception (accessibility); the availability of contraceptive methods and HCPs (availability); the ability to obtain modern contraceptive methods (affordability); and the organisation of services including integration, appointment scheduling, and waiting times (accommodation). Discussions about the acceptability component related more to the quality of care and are reported elsewhere [52].

### Accessibility

Three sub-themes emerged from this theme–geographic access to current contraceptive sources, infrastructure, and preferred source of contraceptive methods.

**Current sources of FP services and contraceptive methods.** Community participants were asked to describe where they could obtain contraception in their community. Public PHCs were identified as the primary source of modern contraception. The primary mode of transport reported across the groups was walking (n = 92), followed by a mixed-use of walking and using a taxi (n = 7) and public taxi (n = 4). Male adolescent participants had the longest travelling time to a source of contraception with a mean travelling time of 86 minutes. Urban female adult participants had the shortest travelling time with a mean time of 13 minutes. Male participants included local shops and PHCs as a major source of modern contraception, while female participants only drew their local PHCs. All participants sketched their

**Table 1. Demographic data for male and female community participants (N = 103).**

| Demographic Characteristics | Community males (n = 25) | Community females (n = 78) |
|---|---|---|
| Age | 23.8 years [15–40] | 26.4 years [15–49] |
| **Educational levels** | | |
| Incomplete secondary school level | 84.0%, n = 21 | 48.7%, n = 38 |
| Completed secondary school level | 16.0%, n = 4 | 51.3%, n = 40 |
| **Relationship status** | | |
| Regular partner, >1yr, not living together | 28.0%, n = 7 | 62.0%, n = 48 |
| Regular partner, <1yr, not living together | 28.0%, n = 7 | 21.0%, n = 16 |
| Regular partner, >1yr, living together | 8.0%, n = 2 | 3.0%, n = 2 |
| Married | 0 | 3.0%, n = 2 |
| Divorced | 0 | 1.0%, n = 1 |
| Casual partner | 16.0%, n = 4 | 0 |
| No current partner | 4.0%, n = 1 | 10.0%, n = 8 |
| Multiple partners | 16.0%, n = 4 | 0 |
| **Pregnancy** | | |
| Reported Pregnancy (N = 61) | 13.1%, n = 8* | 86.8%, n = 53 |
| Unplanned pregnancies | 11.5%, n = 7 | 73.7%, n = 45 |
| Planned pregnancies | 1.6%, n = 1 | 13.11%, n = 8 |
| Average age of first pregnancy [range] | 22 years [20–24] | 19.2 years [17–21] |
| None (N = 42) | 40.5%, n = 17 | 59.5%, n = 25 |
| **Contraceptive method use (N = 86, 83.5%)** | | |
| Total reporting contraceptive use per sex | 25.58%, n = 22 | 74.41%, n = 64 |
| **Per method**$ | N = 22 | N = 64 |
| Male condoms | 77.27%, n = 17 | 75%, n = 48 |
| Depot medroxyprogesterone acetate (DMPA) 3-monthly injection | 27.7%, n = 6* | 29.7%, n = 25 |
| Norethisterone enanthate (NET-EN) 2-monthly injection | 0* | 7.8%, n = 5 |
| Pill | 9.1%, n = 2* | 3.1%, n = 2 |
| Implant | 4.5%, n = 1* | 10.9%, n = 7 |
| Emergency contraception | 18.2%, n = 4* | 1.6%, n = 1 |

*Reports female partner pregnancy/use

$Values do not add to 100%.

experiences accessing contraception, capturing their interpretation of distance and barriers that influence their ability to obtain contraception. (See S1 Table to review data obtained from the drawings and S1 Fig to view examples of the drawings.)

**Infrastructure.** HCP participants described the accessibility of contraceptive methods in relation to the infrastructure where services are provided. The physical size of the PHC determined if a separate room was available where HCPs could provide SRH counselling and contraception.

> P002: At my clinic, there is a room specialising in family planning, but all consultation rooms have family planning. They do it so that there will not be long lines during the day because [the family planning specific room] is closed at 16H00. After 16H00, services continue, and they can still find it in other [general care] rooms. . .
>
> [HCP Group 1, P002]

Space constraints in smaller PHCs facilities resulted in contraception being provided in all consulting rooms.

*P003: We do not have [a specific place if you want family planning] because our clinic is small. But all consultation rooms [have] all services. . .*

*[HCP Group 1, P003]*

### Preferred source of FP services and contraceptive methods

The participants reported that local PHCs, mobile clinics, schools, and private health care were preferred sources of contraception.

One female community participant explained her preference for mobile clinics:

*P001: [I]t could be better if there can be a mobile clinic because sometimes you don't have transport fare to go to the clinic because the clinic is far. Maybe the [Department of] Health can make means [. . .] so that we can have easy access.*

*[Married Female FGD, P001]*

Schools were also identified as sources of contraception. One female adolescent participant noted her preference to obtain her contraceptive method from her school lavatory:

*P010: I would like to get them from school by the toilet (Group laughs). Nobody knows what's happening inside there [. . .]*

*[Rural Female Adolescent FGD, P010]*

Others said they would prefer to access contraception from a private health care provider—especially from doctors. Young adult male participants expressed a preference for male doctors:

*F: Who would [you] like to get [contraception] from?*

*P003: From an experienced doctor who [is qualified for] these things.*

*F: Number 2.*

*P002: Doctor as well.*

*F: Number 1.*

*P001: I also say from the doctor. Especially a male one.*

*F: Male?*

*P001: Because a male [. . .], he is going to understand [. . .] how it works, because a female will [not] tell you that there are different condoms like this and that [. . .]. He will tell you how [they] work because sizes are not the same. He will tell you that if you have this size, this one works like this. So, it [is] better if it is a male doctor.*

*[Young Adult Male FGD]*

### Availability of contraceptive methods and health care providers

**Availability of contraceptive methods.** The participants reported that short-acting and long-acting reversible contraceptive methods were readily available at local PHCs. However,

access to a variety of contraceptive methods, including longer-acting permanent methods (such as sterilisation), was described as inaccessible and unavailable. One participant described the lengthy process involved in accessing female sterilisation:

*P008: Tubal ligation, [it is not available at the clinics] if you want to do it at the clinic it is a long process that I tried [. . .] You first go to the clinic and get a referral letter so that you can go to [the hospital]. When you get to [the hospital], they start afresh and fill in forms and all that. It is a long process.*

*[Married Female Group, P008]*

**Availability of trained HCPs.**   Another sub-theme of availability was the limited availability of trained HCPs.

*P001: [I] can say that they [HCPs]. . . are not enough because it is not good that you walk a long distance to look for help and at the end, you don't get it because [. . .] there is little staff.*

*[Community KI, P001]*

One health KI raised the fact that HCPs are not well trained in providing SRH care, including contraception:

*KI, P007: The rights of the people are violated when it comes to family planning, sexual and reproductive health rights just by not providing nurses who are trained. It's a violation of people's rights.*

*[KI HCP, IDI, P007]*

**Affordability.**   The free availability of contraception from PHCs was positively described to facilitate the use of contraception. A few community participants raised transport costs as affordability-related barriers. Two sub-themes emerged under affordability. The first was an individual user's ability to obtain services, and the second was the financial impact on male partners.

## Economic ability to access FP service and contraceptive methods

The first sub-theme is the user's financial ability to access SRH services of their own choice–referred to as 'individual power' by community participants. The participants described their limited financial situation as constraining their ability to obtain contraception from a provider of their own choice. One female participant explained:

*F: How would you like to get these family planning and contraception? Where would you like to get them?*

*P006: I don't have a choice—it is at the clinic because it is where I have the power to reach, but if I had the strength, I would have loved to get from my doctor. [. . .] But because of power, you are forced to go to the clinic. But condoms you can buy if you have money [..] But like injection and whatnot, you will not be able to buy.*

*[Urban Adult FGD, P006]*

### Financial cost on male partners

Male participants discussed gendered perspectives about the cost of accessing contraception. The financial reliance of female partners on their male partners to obtain their contraception illustrated the limited financial agency that women have in this setting. It also links to the discussions about having 'power' to get their contraception from a provider of their own choice. Young adult male participants explained:

*P004: It costs her; she has to change her method, you see? It is like that my brother, when she changes, it affects you financially, you see, as an "Outjie" (Guy) because she tells you, "my brother [. . .] it is you who is supposed to give her money", to go and buy these things. Why? Because they are not available at our local clinics.*

*[Young adult male FGD, P004]*

Adolescent males further described how gendered expectations related to economic agency could influence the affordability of contraceptives.

*P010: Ones [condoms] that are available for free don't satisfy us.*

*I: They don't satisfy you?*

*P010: When you use them, the lady complains that 'ayi, you look down on her because you are having sex with her using a [free government condom brand]. [. . .] So, you see that thing and then we are forced to buy [brand of condom] [. . .]*

*P005: I also second number 10. He is really right. The minute you take out a [free government condom] on a girl, she gets the impression that you are not the guy she thought you are, you see? It means that you are forced to go and take [paid brand-name condom] so that she can be satisfied that 'hah babe is having sex with me with such a thing' you see.*

*[Adolescent Male FGD]*

### Accommodation

Discussions about the organisation of SRH services, which refers to the accommodative component of access, were raised in three sub-themes: integrated care, accommodating missed visits, and waiting times.

### Integrated care

Discussions surrounding the integration of the FP programme with PHC services raised numerous challenges with organising an efficient service that could accommodate a variety of client needs. Also raised were the challenges with implementing policy at the ground level. The participants reported that integration negatively affected accessing contraception. HCP KIs involved with family planning over several decades explained:

*KI, P005: You know, when family planning was introduced way back in the 1980s, '89. . . up to maybe 2000, we had a service called state health services. At that time, family planning was working very well. But we've since amalgamated all the primary health care services. That is where we actually miss the boat. [Because] in my opinion, if we still maintained that family planning only service, maybe we wouldn't be having as many teenage pregnancies as we are having.*

*[HCP KI, P005]*

Community participants reported that they had good experiences with an NGO supported facility that focused only on SRH services for young adults and adolescents. However, this facility was closed down. The participants outlined the enabling environment that such a specific facility provided:

*P003: Only family planning [services] were there at [name of facility]. Only those for family planning went there [. . .]. It was fast, they closed it, [. . .] And they were fast because you knew that you came for something that does not include anything else, it was a place for people on prevention only. Now they took us back to the clinic [. . .]. That [is] the thing that made me [stop using] 2-monthly [injection] while [I was doing] well [on it].*

*[Females without children FGD, P003]*

An HCP participant echoed this view:

*P004: There were many girls [at the name of the facility] who were doing family planning there. [It] was closed, unfortunately, [and the clients] had to go to the clinic [. . .] But we lost a lot of girls. They stopped coming to be absorbed at the clinic because it was just for the youth only.*

*[HCP Group 1, P004]*

## Accommodating missed visits

The participants pointed out that PHCs cannot accommodate the needs of a variety of clients, such as those who are employed. This influenced clients' ability to attend PHCs on specific days, especially if they are employed resulting in missed visits:

*P004: Sometimes, there are compelling issues that make you miss your clinic date. [. . .] then you find a temporary job. It is very hard to ask that you [can have time off to] go to the clinic.*

*[Rural Adult Female FGD, P004]*

The participants further reported that the consequences of missing an appointment date were often met with harsh treatment by the nurses:

*P001: [The] things that they do, the nurses, if you have missed your date, [is that] they chase you away, while you were waiting3 hours and above. She will tell you to go and wait outside. You will be the last to be attended [to], that is a turn-off.*

*[Rural Adult Female FGD, P001]*

## Waiting times

Long waiting times were a much-discussed topic that participants related to the QoC, the accommodative ability of a PHC, and as a factor that contributed to their economic burden to obtain contraception. Community participants described having to wait for several hours and that long waiting times could lead to the discontinuation of contraceptive use.

*P003: Yes, I also left it [referring to method discontinuation] because of that, the 2-months [injection], it was treating me well [. . .]. The thing that made me stop the 2-months [injection] is that thing of the nurses. They sometimes attend to you and [sometimes] they do not attend to you [. . .]. You sit there the whole day at the clinic. You will arrive in the morning at ten, eleven. You will leave at four not [being] attended to. It's just busy. What's worse is they mix you with the sick [clients] sometimes while they said, there is a place that used to be called 'be injected' [the specialised NGO supported FP service that was closed].*

*[Female Group without Children, P003]*

One female participant explained that even though there may only be a few clients for contraception, they still had to wait lengthy times:

*P001: Because we sit more than three hours at the clinics when you're there to take the injection. Sometimes, you find four [clients only] who are there for contraceptives.*

*[Rural Adult Female FGD, P001]*

HCPs explained the long waiting times and felt that it impacted the quality of care:

*P005: [O]ur clients wait for a long time to be seen because they need to be taught about family planning. Then from there, they come back to the queue. So, it's not good quality [service] for the clients [. . .].*

*[HCP Group 1, FGD, P005]*

Another problematic factor raised about time was that in some PHCs, adolescent school children were attended to before adult clients, resulting in tension and services not being accessed:

*P004: They usually start with children, yet you are also in a hurry to get to your work as a domestic worker in the suburbs [. . .]. You end up taking your card, putting it in your bag and leaving.*

*[Rural Adult Female FGD, P004]*

## Discussion

In this paper we explored access to SRH services and modern contraception from a community and HCP perspective. Access to healthcare services is a multidimensional concept [33, 34, 40]. This makes the evaluation of access complex as numerous definitions are used. Pench- ansky and Thomas [34] define access as "the general concept which summarizes a set of more specific areas of fit between the patients and the health care system" (p.128). Aday and Ander- sen [53] suggest that access is a multidimensional concept that can be viewed as either poten- tial or realised and defined as "those dimensions which describe the potential and actual entry of a given population group to the health care delivery system" (p.51). More specifically to SRH and contraceptive services, Bertrand et al. (1995) [37], define access as "the degree to which family planning services and supplies may be obtained as a level of effort and cost that is both acceptable to and within the means of a larger majority of the population" (p.65). All these definitions highlight the importance of resources being available, but also that other cru- cial factors such as the society, geographical access, capacity of the healthcare service, cultural

beliefs, and socio-cultural factors are considered when discussing access. Access therefore extends beyond the mere availability of facilities [40].

The community perspective included users and non-users of contraception that allowed a robust discussion about the barriers to accessing contraception. The HCPs provided context and insight into the daily challenges that they face in delivering contraception. These various perspectives broadened current understanding about what access to contraception means in this setting, beyond the narrow categories used in the SADHS (2016) data. The overall results showed that the accommodation component of access posed the biggest problems for community members to access SRH services, and HCPs to provide care. In addition, personal agency and choice in service delivery point (SDP) and provider played significant roles in peoples' ability to access contraception.

Penchansky and Thomas's (1981) access framework which includes five distinct, but inter-related components were used to explore the data. Categorising the inductive emergent themes and sub-themes according to each component of access allowed for a robust but specific exploration about access to contraception. This robust understanding could not have been achieved if more recently proposed frameworks, such as those proposed by Bertrand et al. [37] or Choi et al. [33], or the AAAQ were used.

The influence of policy and the implementation of policy on accessing contraception was seen throughout the results. Even though SA has progressive and enabling SRH policies the implementation of these policies was found to be problematic. This finding has been reported previously [16], and its persistence over time shows the difficulty in drafting relevant policies and their implementation at the ground-level. Challenges in implementing policies resulted in the misinterpretation of access and availability to contraceptive services by community members. The misunderstanding of policy implementation points to the need for greater engagement and communication between community members and HCPs [54].

Discussions about the integration of SRH services with general PHC services demonstrated the challenges with implementing policies. Restructuring the specialised vertical FP programme to integrate with PHC was some of the most significant policy and service implementation changes made to the delivery of modern contraception in SA [16, 17, 25, 55]. Despite almost two decades of attempting to integrate FP services with district based PHC services, full implementation is still lagging, yet the importance and need for successful integration are evident [16, 30, 56, 57]. In contrast to international findings however, results from this study showed that integration negatively affected access to contraception in this setting [58]. This finding adds to explain the report by Adeniyi et al [57] that unintended pregnancies remained high in this setting despite the integration of services. Furthermore, buy-in from HCPs is key to the successful implementation of integrated services [15, 16]. HCP participants from this study reported their frustrations with providing integrated SRH care and their preference for a specialised service. This was echoed by reports from community members who voiced their dissatisfaction about integrated services that led to overcrowded facilities. Overcrowded and overburdened facilities resulted in long waiting times which contributed to discontinuation of contraceptive use. McIntyre and Chow [59] found that waiting times were often misreported. Yet, lengthy waiting times was the most significant sub-theme reported in this data related to access to care. While national policy state that waiting times at PHCs should average around two hours, participants from this study reported waiting four hours or more just to receive contraception further illustrating the discord between policy and policy implementation at the service level [60]. Addressing long waiting times in PHCs is a complex problem that will require adjustments in other essential areas such as having sufficiently trained HCPs, ensuring that capacity is not over-extended, and adjusting the operating hours of PHCs. Currently, no model of integration effectively deals with this important problem.

Other reasons cited that contribute to lengthy waiting times included restrictive operating hours and the limited availability of trained HCPs. The limited operating hours of PHCs (08:00 am to 16:00 pm) restricted access to contraception. In addition, there are reports that in some South African PHCs, nurses only attend to clients in the mornings, reserving afternoons for administrative work, thereby further limiting operating hours [17]. Employed participants reported that these operating hours were not compatible with their work hours resulting in them either losing income due to a loss of working hours or missing their appointment visit resulting in discontinuation of their contraception. Scheduling appointments is one strategy to improve accommodation within PHCs, however missed visits are common in this setting due to the reasons stated. Baumgartner et al [61] reported that nearly 50% of Depomedroxyprogesterone (DMPA) users return late for their follow-up appointment, with some late returning clients denied access to follow-up DMPA injections. The accommodation component, which included discussions about integration, long waiting times, and operating hours, was therefore identified as having the most significant influence on accessing contraception.

The limited availability of trained HCPs was discussed under the availability theme. Following the restructuring of the health care system was the shift in the training of nurses from specialised skills, such as family planning and contraception, to generalised skills [62]. Adequately trained HCPs are essential to expanding access to contraception, but specialist trained nurses have fallen away [15, 63]. The lack of trained HCPs was raised as a concern in this data, and while staff shortage is not a new finding [64, 65], it highlights the need for human resources to be included in discussions about the availability of contraception.

While the widespread availability of short-acting contraceptive methods was positively associated with access, the difficulty to access long acting or permanent contraceptive methods were negatively reported–a finding supported by the literature [5, 6]. The female participants reported dissatisfaction with the unattainability of long-acting and permanent methods such as sterilisation.

Data under the affordability and accessibility themes revealed that individual agency and choice in provider and service delivery point played important roles in accessing SRH services.

SRH services and contraception is provided free of charge in the public sector and the participants positively associated this with accessibility to contraception. However, the data showed that the free availability of contraception was constrained by an individual's economic power and ability to obtain contraception from an SDP and provider of their choice. Discussions about affordability extended beyond the usual reports about the price of services and methods. Instead, the data from the drawings highlighted individual agency in accessing contraception and that although the methods are freely available there is a cost to obtaining them. Women reportedly relied on their male partners for economic support to reach PHCs. Male participants added an interesting perspective by drawing a shop in addition to a PHC. Men possibly constructed their ability to access contraception in this way because they have the economic means to pay for privately sourced contraception such as branded condoms. Women captured their limited agency by only drawing PHCs as a possible SDPs. Even young adolescent females only identified PHCs, instead of other sources such as schools. Limited female agency also reflects the unequal gendered economic agency present in these communities.

Further to the discussions of agency was the importance of choice in SDP and provider. Penchansky and Thomas [34] showed that having the same HCP of choice is essential for retention in care and satisfaction with available services–a particularly important point for continued contraceptive use. As explained by the participants, in the SA public health district based PHC model, most women have limited autonomy to choose an SDP from which to obtain contraception. Unless they have the economic means to obtain contraception from

private health care facilities, which for most South Africans is unaffordable, their only option is their local PHC [32].

Lastly under the accessibility theme, HCPs discussed the impact of infrastructure on accessing FP services. HCPs reported that PHCs with insufficient space or adequate infrastructure constrained access to contraception. Ailing infrastructure also has a negative effect on the psychological well-being of HCPs, further limiting access to services [16, 62].

## Limitations and strengths

Some limitations exist for this study. Firstly, the data presented here are only based on experiences with the public health care sector. For a more comprehensive comparison and understanding of access to contraception, data and experiences from private health care users should also be evaluated. Secondly, as this was a qualitative study, the results are not necessarily generalisable. Instead, the focus is on the transferability of the methods and findings to other settings. While the findings are specific to the South African context, specifically KZN, the methods for evaluating access using a holistic approach and identifying the relevant components are relevant for other settings. The selection of the community and healthcare participants into this study was from two areas—one urban and the other being peri-urban/rural. Participants were also selected on specific criteria, such as marital/partner status and whether they had children or not. Such specific criteria could potentially have resulted in biased opinions as some members of other communities or those in other types of relationships could have varying opinions. An area where bias was reduced was that non-contraceptive users were also included in this study, which balanced the view of contraceptive access and use, especially in terms of non-use.

Using a community perspective adds to this paper's contribution to the literature. The view of facility-based clients is usually used to evaluate satisfaction with services. These types of reports can be skewed toward positive overreporting out of fear of repercussions [66]. Interviewing community members who may or may not be FP clients allows for a less-influenced perspective and evaluation on accessing FP services.

A few questions about accessing FP services remained unanswered in this data set and required further exploration. Adolescent males reportedly had the longest travel time to SDPs, but it was not well explained in the data as to why. Female participants did not identify any other sources for contraception other than PHCs–this needs exploration. Long waiting times are a significant issue that requires urgent further investigation to improve access to services.

## Conclusion

This paper sheds light on how community members and HCPs evaluate access to SRH services and contraception. In contrast to the results presented by the SADHS (2016), the data showed that significant barriers to accessing contraception exist in this setting which can result in contraceptive discontinuation. Overall, most of the female participants were satisfied with the accessibility and affordability components. They were less satisfied with the availability of long acting and permanent contraceptive methods. The limited number of trained providers was raised as a concern. Accommodation was identified as the component that presented the most significant barriers to accessing SRH services. In particular, the ineffective organisation of services resulted in lengthy waiting times that could result in the discontinuation of contraceptive use. The addition of the male perspective in accessing contraception was a new finding on this subject. Men described the financial burden of accessing contraception and highlighted the importance of having an alternative source of contraception in the communities that are easily accessible.

The importance of recognising that access is a multidimensional concept and that the most important components should be identified to understand the challenges in accessing contraception was made clear. Careful attention must also be paid to policy implementation over time, as local contextual changes could make policies redundant. Policies must be evaluated for their effectiveness over time. Women face many obstacles when accessing [6] contraception. The argument here is that the health care system itself should not be an obstacle. Every effort should be made to ensure that all women who need or want contraception should be able to get it without much hassle or having to overcome numerous barriers.

Even though the data for this study were collected before the onset of the COVID-19 pandemic, the findings highlight the importance of paying attention to all the access components. Health care facilities and systems must identify where access problems exist and address them appropriately to be prepared for unexpected new epidemics and challenges.

## Supporting information

**S1 Data. IDI guide.**
(PDF)

**S2 Data. FGD female guide.**
(PDF)

**S3 Data. FGD male guide.**
(PDF)

**S4 Data. FGD health care providers guide.**
(PDF)

**S1 File.**
(PDF)

**S1 Table. Data from the drawings.**
(DOCX)

**S1 Fig. Participant drawings of distance to source of contraception.**
(TIF)

## Acknowledgments

The authors would like to thank all the participants who gave their valuable time to participate in this study. We would also like to thank the UPTAKE team who collected the data and assisted with transcribing and translating the audio.

## Author Contributions

**Conceptualization:** Joanna Paula Cordero, Petrus S. Steyn, Jennifer Ann Smit.

**Data curation:** Yolandie Kriel, Cecilia Milford, Joanna Paula Cordero.

**Formal analysis:** Yolandie Kriel, Cecilia Milford.

**Funding acquisition:** Joanna Paula Cordero, Petrus S. Steyn, Jennifer Ann Smit.

**Investigation:** Yolandie Kriel, Cecilia Milford, Joanna Paula Cordero, Petrus S. Steyn.

**Methodology:** Cecilia Milford, Joanna Paula Cordero, Petrus S. Steyn, Jennifer Ann Smit.

**Resources:** Jennifer Ann Smit.

**Validation:** Yolandie Kriel, Cecilia Milford.

**Writing – original draft:** Yolandie Kriel.

**Writing – review & editing:** Yolandie Kriel, Cecilia Milford, Joanna Paula Cordero, Fatima Suleman, Petrus S. Steyn, Jennifer Ann Smit.

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
