## [Decision Letter · Decision Letter 0]

4 Jan 2022

PONE-D-21-18604Access to public sector family planning services and modern contraceptive methods in South Africa: A qualitative evaluation from community and health care provider perspectives.PLOS ONE

Dear Dr. Kriel,

Thank you for submitting your manuscript to PLOS ONE. After careful consideration, we feel that it has merit but does not fully meet PLOS ONE’s publication criteria as it currently stands. Therefore, we invite you to submit a revised version of the manuscript that addresses the points raised during the review process.

As indicated by the reviewers, additional detail is needed regarding the drawings - why they were used, the value added, and how they were analyzed. Furthermore, copy editing and removal of redundant information should be addressed before submitting a revised version of the paper. Please submit your revised manuscript by Feb 18 2022 11:59PM. If you will need more time than this to complete your revisions, please reply to this message or contact the journal office at plosone@plos.org. Please include the following items when submitting your revised manuscript:A rebuttal letter that responds to each point raised by the academic editor and reviewer(s). You should upload this letter as a separate file labeled 'Response to Reviewers'.A marked-up copy of your manuscript that highlights changes made to the original version. You should upload this as a separate file labeled 'Revised Manuscript with Track Changes'.An unmarked version of your revised paper without tracked changes. You should upload this as a separate file labeled 'Manuscript'.

We look forward to receiving your revised manuscript.

Kind regards,

Funmilola M. OlaOlorun, PhD

Academic Editor

PLOS ONE

Journal Requirements:

2. Please include a copy of the interview guide used in the study, in both the original language and English, as Supporting Information, or include a citation if it has been published previously.

4. We noted in your submission details that a portion of your manuscript may have been presented or published elsewhere. Table 1 ,2 , 3 and 4 Please clarify whether this  publication was peer-reviewed and formally published. If this work was previously peer-reviewed and published, in the cover letter please provide the reason that this work does not constitute dual publication and should be included in the current manuscript.

7. We note you have included a table to which you do not refer in the text of your manuscript. Please ensure that you refer to Table 4 in your text; if accepted, production will need this reference to link the reader to the Table.

Additional Editor Comments (if provided):

This is an interesting paper with great potential. The two reviewers have provided detailed feedback that can be used to improve the paper further. Please review these comments carefully as you work on a revision.

Reviewers' comments:

Reviewer's Responses to Questions

**Comments to the Author**

1. Is the manuscript technically sound, and do the data support the conclusions?

Reviewer #1: Yes

Reviewer #2: Yes

2. Has the statistical analysis been performed appropriately and rigorously? 

Reviewer #1: N/A

Reviewer #2: N/A

3. Have the authors made all data underlying the findings in their manuscript fully available?

Reviewer #1: No

Reviewer #2: No

4. Is the manuscript presented in an intelligible fashion and written in standard English?

Reviewer #1: Yes

Reviewer #2: Yes

5. Review Comments to the Author

Reviewer #1: This is an interesting paper, well-written and comprehensive. However, it is also a very long paper!!! The authors have unnecessarily described the context and background as though it is a mini thesis which much duplication as well. I find the detailed description of the frameworks repetitive- perhaps more than sufficient in the methods but not that necessary in the background as this is not a thesis... detailed comments below

Introduction

- I would recommend the authors to integrate and shorten the background and introduction pieces together, the depth of information is quite overwhelming, which takes kills the enthusiasm for the findings of this study-the main point of this paper, in my view.

Methods

- The first 2 paragraphs under setting seem to belong in the introduction/background section too. Other than that, the methods are well-described!

- In Table 1, I'm unclear about the sample groups from number 7-12: are these neither from rural/urban, which setting do they come from? are they exclusive from the sample groups 1-6? in other words, are the sample groups 1-6 not in a relationship? not married or single? have children???? the classification/categorization is quite confusing...e.g. the authors included young females from rural and urban, 20-34 yrs old, and also have the same age group single, married/in a relationship, and with no children (18-49 yrs)- are they different from the ones described in 1-6 and if so, how?

- Only the applicable theoretical framework should be described in the methods section, concisely as well.

Results

-Very comprehensive and interesting findings! In general, I like how they are presented-following on the framework structure and flow logically.

- The theme on implementation policy is very thin- I'd consider that part in the discussion section rather than a stand alone theme..

- Insert figure 2 has nothing to do with clinic, only a shop shown there.

Discussion

- Nice discussion as well, especially the policy part! However, I would suggest that is framed in a "policy implication" perspective after discussing the above findings.

- Perhaps no need for the "theoretical framework" structure here- just discuss the findings!

- It is also relatively long!

- Notwithstanding the framework within which this paper was conceived on, it is surprising to see the most recent "research" related reference used by the authors from 2019- surely more and relevant research has been published between 2019 and 2021 in relation to access, uptake and use of modern contraceptives. So I would challenge the authors to revisit the literature and update their references...

Reviewer #2: General

This paper is interesting and has a rich data set. Overall some of the data needs to be reviewed and expanded in the findings and discussion to adequately apply a social constructionist framework. The paper needs a minor grammar edit and a review of spelling. See abstracts in annotated paper review for a few examples.

Introduction

The section on the critique of the access frameworks needs to be tighter (lines 72-81). See comments on paper.

Methods

The methods should more fully discuss the use of drawings; why they were used, how were they set up and used in the facilitation of the FGDs, and how were they analysed in relation to the coded transcripts. Were they only used to comment on access to services and geographical barriers?

Findings

The finding related to policy is weak. What more did the data say? How was integration of services experienced by women? This is discussed later but could also be explored or at least cross-referenced as it reflects the reality between policy and implementation.

The use of the drawing as a basis for Table 5 and then statistics related to the drawings seems to me a rather static way of using the drawings – would you have not got the same data had these been asked in the focus groups? Were there other things that people drew that supplemented the data and helped with a deeper understanding (social construction of meaning).

See other comments on marked up paper for specific areas that need strengthening.

6. PLOS authors have the option to publish the peer review history of their article (what does this mean?). If published, this will include your full peer review and any attached files.

Reviewer #1: **Yes: **Kim Jonas, PhD

Reviewer #2: No

---

## [Author Response · Author response to Decision Letter 0]

3 Jun 2022

Response to reviewers

Thank you to the reviewers and the editor for your comments. We have addressed all the issues that were raised and responded to your suggestions as we deemed appropriate. 

1. Please ensure that your manuscript meets PLOS ONE's style requirements, including those for file naming. The PLOS ONE style templates can be found at https://journals.plos.org/plosone/s/file? id=wjVg/PLOSOne_formatting_sample_main_body.pdf and https://journals.plos.org/plosone/s/file? id=ba62/PLOSOne_formatting_sample_title_authors_affiliations.pdf

- Response:

Thank you, the formatting was updated as per the journal style. 

2. Please include a copy of the interview guide used in the study, in both the original language and English, as Supporting Information, or include a citation if it has been published previously.

- Response:

The interview guides are included now as part of the supporting information.

- Response:

The funding information will be updated.

4. We noted in your submission details that a portion of your manuscript may have been presented or published elsewhere. Table 1 ,2 , 3 and 4 Please clarify whether this publication was peer-reviewed and formally published. If this work was previously peer- reviewed and published, in the cover letter please provide the reason that this work does not constitute dual publication and should be included in the current manuscript.

- Response:

The tables were removed to reduce any dual publication. Careful attention was paid to avoid any dual publication among the manuscripts that arise from this dataset. Some information like the demographic data might be similar due to the nature of such data. All other text and findings presented are unique to each paper. 

As explained in the original cover letter and submission this paper forms part of a larger project, The UPTAKE Project, and the main author’s PhD thesis. 

5. We note that you have indicated that data from this study are available upon request. PLOS only allows data to be available upon request if there are legal or ethical restrictions on sharing data publicly. For more information on unacceptable data access restrictions, please see http://journals.plos.org/plosone/s/data-availability#loc-unacceptable-data- access-restrictions.

- Response

The database is not publicly available as it contains information that could compromise research participants' privacy and consent. However, some anonymised aspects of the datasets may be available upon request and with the permission of the Department of Sexual and Reproductive Health and Research, World Health Organization, and the MatCH Research Unit (MRU). Note that data sharing is subject to WHO data sharing policies and data use agreements with the participating research centre, MRU. For permission to access the database, please contact the MatCH Research Unit at info@mru.ac.za

b) If there are no restrictions, please upload the minimal anonymized data set necessary to replicate your study findings as either Supporting Information files or to a stable, public repository and provide us with the relevant URLs, DOIs, or accession numbers. For a list of acceptable repositories, please see http://journals.plos.org/plosone/s/data- availability#loc-recommended-repositories.

- Response:

Thank you, the ethics statement was removed from the statements section. 

7. We note you have included a table to which you do not refer in the text of your manuscript. Please ensure that you refer to Table 4 in your text; if accepted, production will need this reference to link the reader to the Table.

- Response:

Thank you for noting this. This table was removed to improve the word count of the manuscript and reduce any dual publication. 

- Response:

Captions for the supporting files were included at the end of the manuscript according to the PLOS guideline. 

Additional Editor Comments (if provided):

This is an interesting paper with great potential. The two reviewers have provided detailed feedback that can be used to improve the paper further. Please review these comments carefully as you work on a revision.

Reviewer #1: This is an interesting paper, well-written and comprehensive. However, it is also a very long paper!!! The authors have unnecessarily described the context and background as though it is a mini thesis which much duplication as well. I find the detailed description of the frameworks repetitive- perhaps more than sufficient in the methods but not that necessary in the background as this is not a thesis... detailed comments below.

- Response:

Thank you for your review and comments. After careful revision, I have tried to remove all duplication where I could see it. I have also edited the literature/background sections and discussion to reduce the word length and address the reviewers’ comments. 

Introduction

- I would recommend the authors to integrate and shorten the background and introduction pieces together, the depth of information is quite overwhelming, which takes kills the enthusiasm for the findings of this study-the main point of this paper, in my view.

- Response:

Thank you, this was done. The literature and background sections were edited to be one section. 

Methods

- The first 2 paragraphs under setting seem to belong in the introduction/background section too. Other than that, the methods are well-described!

• Response:

This was edited as per suggestion. 

- In Table 1, I'm unclear about the sample groups from number 7-12: are these neither from rural/urban, which setting do they come from? are they exclusive from the sample groups. 1-6? in other words, are the sample groups 1-6 not in a relationship? not married or single? have children???? the classification/categorization is quite confusing...e.g. the authors included young females from rural and urban, 20-34 yrs old, and also have the same age group single, married/in a relationship, and with no children (18-49 yrs)- are they different from the ones described in 1-6 and if so, how?

• Response:

I have removed the table and written the group descriptions to help clarify. To clarify the participant breakdown – in total 127 participants participated in this study in SA. Of the 127 participants, 16 were health care providers and 8 in each FGD which were divided according to professional ranking. 

There were also 8 key informant IDIs.

The female community participants were first divided between the urban and rural settings. Three urban groups consisted of women aged 35 to 49 years (adults), 20 to 34 years (young adults), and 15 to 19 years (adolescents). These three groups were matched for age in the rural setting. 

Three additional FGD groups were then held with a group of unmarried women (20 to 34 years), a group of married/in-union women (20 to 34 years), and a group of women who do not yet have children (18 to 49 years). These three groups were recruited from both the rural and urban areas. 

All the FGD participants were unique for each group i.e., no participant was interviewed twice. 

The 3 male FGD groups consisted of men aged 35 to 49 years (adults), 20 to 34 years (young adults), and 15 to 19 years (adolescents) respectively and were a mix from the rural and urban areas. 

- Only the applicable theoretical framework should be described in the methods section, concisely as well.

• Response:

Thank you, this section was edited to only refer to the applicable theoretical framework. After revising the manuscript, I noted how this can be confusing for some readers. 

Results

-Very comprehensive and interesting findings! In general, I like how they are presented - following on the framework structure and flow logically.

- The theme on implementation policy is very thin- I'd consider that part in the discussion section rather than a stand-alone theme.

• Response:

Thank you for this comment. Based on your review and the comment from reviewer 2, I decided to remove policy implementation as a theme and to rather discuss it under the discussion section. 

- Insert figure 2 has nothing to do with clinic, only a shop shown there.

• Response:

Yes, some male participants only drew shops as their source of contraception. This drawing was included to show this finding. 

Discussion

- Nice discussion as well, especially the policy part! However, I would suggest that is framed in a "policy implication" perspective after discussing the above findings.

- Perhaps no need for the "theoretical framework" structure here- just discuss the findings!

- It is also relatively long!

• Response:

Thank you for the suggestions made. On revision, I have edited the discussion to reduce the length and make it more concise. Please see the update discussion section. 

- Notwithstanding the framework within which this paper was conceived on, it is surprising to see the most recent "research" related reference used by the authors from 2019- surely more and relevant research has been published between 2019 and 2021 in relation to access, uptake and use of modern contraceptives. So I would challenge the authors to revisit the literature and update their references...

• Response:

Thank you for this comment. On review, there really is very limited relevant literature about access to contraception in SA – and none which are more recent in publication date. This highlights the importance of this paper as an update to what was published in the past. I also wanted to keep the word length shorter as the paper is already quite long, so I only included literature that is absolutely relevant to this paper. 

Reviewer #2: General

This paper is interesting and has a rich data set. Overall some of the data needs to be reviewed and expanded in the findings and discussion to adequately apply a social constructionist framework. The paper needs a minor grammar edit and a review of spelling. See abstracts in annotated paper review for a few examples.

• Response:

Thank you for the review of the paper. I apologize for the grammatical errors which were updated and corrected. I have revised the results and discussion section to bring out the social constructionist stance, but I am constrained by an already lengthy manuscript to fully explore this issue. In addition, the social constructionist stance can also be appreciated from a methodological point of view, as both health care providers, key informants, and community participants’ voices were included in the findings. This allowed for each grouping, especially the divide between the health care providers and community members, to be elucidated without singling out which was more important. 

Please see the track change version of the paper for the edits and corrections made. 

Introduction

The section on the critique of the access frameworks needs to be tighter (lines 72-81). See comments on paper.

• Response:

Thank you for your comments. I have updated the theoretical framework section. Please see the manuscript for the updated text. 

Methods

The methods should more fully discuss the use of drawings; why they were used, how were they set up and used in the facilitation of the FGDs, and how were they analysed in relation to the coded transcripts. Were they only used to comment on access to services and geographical barriers?

• Response:

Thank you for the comment. The drawings were an additional method to collect individual data within the group setting. Using this method was quite effective in allowing individual participants to share their views on accessing contraception. Overall, the drawings were quite simple and reflected what was asked: where their source of contraception was, how they got there, and how long it took. 

The participants enjoyed making the drawings - it served as a bit of a break in the FGD since these FGDs were rather long. 

I have added some text in the methodology section that expands a bit on the drawings. The drawings were analysed in the same manner as the rest of the data – the codebook was used, and appropriate codes were applied. These codes mainly consisted of accessibility codes, such as distance, source of contraception, time travelled, and mode of transport. 

Findings

The finding related to policy is weak. What more did the data say? How was integration of services experienced by women? This is discussed later but could also be explored or at least cross-referenced as it reflects the reality between policy and implementation.

- Response:

Based on this comment and the comments from reviewer 1 I decided to remove policy as a theme and rather bring it up in the discussion. Policy and implementation truly affected all the areas of access and were challenging to isolate. I attempted to integrate policy discussions throughout the discussion and especially to highlight how integration impacted women. 

The use of the drawing as a basis for Table 5 and then statistics related to the drawings seems to me a rather static way of using the drawings – would you have not got the same data had these been asked in the focus groups? Were there other things that people drew that supplemented the data and helped with a deeper understanding (social construction of meaning).

- Response:

A key challenge with using a visual data collection technique in qualitative research is the question of whether the same results could not be obtained by a different method. 

According to the literature, the only true way to do that is to set up a rigorous experiment. For this study, the aim was not to prove the validity of one method over another but rather to obtain an updated perspective on accessing contraception. Most of the data were obtained from the interview transcripts but the drawings allowed for an individual data collection method in a group setting. This eliminated the need to repeat the data collection process. It also served as a ‘break’ during the discussions allowing the participants to interact freely with the interviewers and each other. 

Could the same data have been obtained during the FGD itself by merely asking the question about how they experienced accessing contraception? Perhaps, but the individual experience would have been lost. Each participant drew their own experience, and their own specific challenges when going to collect their contraception. That level of detail would have been lost in a group discussion since not every participant responds to every question asked. It was important for this study to gain an understanding of how, where, and what challenges participants experienced when they wanted to access contraception. This was especially the case because not all the participants were current or ever contraceptive users and we wanted to understand where the barriers may lie. 

See other comments on marked-up paper for specific areas that need strengthening.

- Responses to comments in PDF:

- All grammar and spelling errors were corrected. 

- Lines 72 to 81 – moved to the theoretical framework section to improve the flow of the paper. The critique was emphasized in the updated section. 

- Lines 137 – 138 – Sentence was updated as suggested.

- Lines 186 – 188 - Thank you for the comment. The drawings were an additional method to collect individual data within the group setting. Using this method was quite effective in allowing individual participants to share their views on accessing contraception. Overall, the drawings were quite simple and reflected what was asked: where their source of contraception was, how they got there, and how long it took. 

The participants enjoyed making the drawings - it served as a bit of a break in the FGD since these FGDs were rather long. 

I have added some text in the methodology section that expands a bit on the drawings. 

- Line 296 – Thank you for the comment. As per the reviewers’ suggestion, I have decided to remove the policy theme from the thematic results and to rather discuss it in the discussion since the influence of policy and policy implementation was so pervasive in the data set.

- Line 433 – 434 – Thank you for the comment. The sentence was edited to include the reviewer’s suggestion. 

- Line 452 – 456 – In the results, the participants explain that the focus on the family planning services is lost due to the clinics being overburdened with a variety of clients. Both the community members and the health care providers noted this. This topic was elaborated on a bit in the updated discussion. 

- “Your data also shows that integration of services at PHC level (policy directed) has possibly (unintentionally) resulted in lower levels of access- as you describe waiting times, missed visits etc etc” – Thank you for the comment. This was highlighted in the discussion.

- Lines 600 – This sentence was in reference to the literature that was cited. The sentence was removed for clarity. 

- Lines 616 – 621 - Thanks for this comment. I agree that the issues of agency, autonomy and economic power play an important role in contraceptive use (or non-use). The discussion was updated to reflect these important points. 

- Lines 635 – 640 - Thank you for the comment. We agree this reflects the limits of policy implementation and have added a sentence to help clarify.

---

## [Decision Letter · Decision Letter 1]

2 Aug 2022

PONE-D-21-18604R1Access to public sector family planning services and modern contraceptive methods in South Africa: A qualitative evaluation from community and health care provider perspectives.PLOS ONE

Dear Dr. Kriel,

Thank you for submitting your manuscript to PLOS ONE. After careful consideration, we feel that it has merit but does not fully meet PLOS ONE’s publication criteria as it currently stands. Therefore, we invite you to submit a revised version of the manuscript that addresses the points raised during the review process.

We look forward to receiving your revised manuscript.

Kind regards,

Funmilola M. OlaOlorun, PhD

Academic Editor

PLOS ONE

Additional Editor Comments (if provided):

Thank you for working on the feedback you received to improve your manuscript. There are still a few issues that need to be clarified, and explained such as not to cause confusion. Please pay close attention to the additional feedback from your reviewers, particularly reviewer 3. In order to be suitable for publication, it is important that your language and choice of terminologies are very clear. Thank you for attending to this.

Reviewers' comments:

Reviewer's Responses to Questions

**Comments to the Author**

1. If the authors have adequately addressed your comments raised in a previous round of review and you feel that this manuscript is now acceptable for publication, you may indicate that here to bypass the “Comments to the Author” section, enter your conflict of interest statement in the “Confidential to Editor” section, and submit your "Accept" recommendation.

Reviewer #3: (No Response)

Reviewer #4: (No Response)

2. Is the manuscript technically sound, and do the data support the conclusions?

Reviewer #3: Partly

Reviewer #4: Yes

3. Has the statistical analysis been performed appropriately and rigorously? 

Reviewer #3: N/A

Reviewer #4: N/A

4. Have the authors made all data underlying the findings in their manuscript fully available?

Reviewer #3: No

Reviewer #4: No

5. Is the manuscript presented in an intelligible fashion and written in standard English?

Reviewer #3: Yes

Reviewer #4: Yes

6. Review Comments to the Author

Reviewer #3: This paper attempts to fill a gap which is the lack of use of modern contraceptive methods in KwaZulunatal, South Africa. The paper is well written and qualitative data shed some light on “access” to family planning methods from a broader perspective that include five elements according to a theoretical framework. However, these five elements needs to be well defined from the start. Unfortunately, data don’t bring much new insights in particular on the acceptability issue including the issue of secondary effects which is reported worldwide as the main reason for lack of uptake or continuation. The social factors or what is called the “social constructionism” lens used to view the data could be more prominent and data presented and discussed in light of social determinants (unmarried or single women or other vulnerable groups).

1. There is some confusion or contradictions along the text with the term “access” despite the fact that you present a framework you will use in this study and you define access accordingly.

It should clearly presented from the start in the theoretical framework section after defining access, what each term (Five interrelated components of access used in this framework, namely accessibility, availability, affordability, accommodation, and acceptability) will entail, stick to that and be consistent.

Page 2 lines 47-48 it is mentioned :” This study illustrated the importance of examining access as a holistic concept and not just the availability of services or methods.” I don’t think that access is commonly related or limited to the availability of contraceptive methods; on the contrary access refers primarily to geographical and financial access. I would remove or reformulate that sentence.

Moreover in the introduction page 4 lines 96-100: you had a contradictory statement as a justification of the paper: “Only focusing on affordability and physical access contributes to an inaccurate understanding of access. Other sources have identified additional barriers, including poor quality of care (QoC), limited variety of modern contraceptive methods, fragmented integration of sexual and reproductive health (SRH) services with PHC services, ailing infrastructure, shortage of trained nurses, poor attitudes of nurses, and stigma (5, 15-19)”

Results : Line 280 Accessibility: “Three sub-themes emerged from this theme – geographic access to current contraceptive sources, infrastructure, and preferred source of contraceptive methods.” Then later line 296 “ Infrastructure : HCP participants described the availability and accessibility of contraceptive methods at their healthcare facility related to the available infrastructure.

Again in the Conclusion there is some contradiction and confusion regarding the meaning of access: It is stated line 616 “Overall, the participants were satisfied with the accessibility and affordability components” and later line 623 “ Men described the financial burden of accessing contraception and highlighted the importance of having an alternative source of contraception in the communities that are easily accessible.” Please clarify and reformulate

Please define the terms from the start and harmonize the text and the abstract.

2. Page 5 : the second paragraph (Lines 105-120) is long and could be quite shortened if you start the paragraph by the following : “Despite the enabling policies and restructuring, the SA public health sector has struggled to 118 make real gains in improving access to services, including FP service (20, 29, 30).

3. Introduction: Although your study focuses on the public sector, it would be indicated to mention in the introduction (and also in the settings- eThekwini District of KwaZulu-Natal SADHS results related to what are the main sources of family planning in SA, in particular: % of public vs. private services? , and most preferred sources of contraception among young people and adults ? and most contraceptive methods used in SA and KwaZulu-Natal.

4. Page 8 “CPR, “for KZN has consistently fallen below the national median, with the 2018/19 CYPR 154 being 44.4%” then later “The CYPR for eThekwini is also below the national average 159 at 44.4% for 2018/2019” please clarify for consistency.Page 8 lines 155: Study settings: Please add HIV prevalence (then the total population affected by HIV) in KwaZulu natal.

5. Page 8 line 171 “Twelve of these FGDs were with community participants (n= XX) , and two with HCP (n= XX) ” I think you removed the table (duplication of data presentation) but please mention the total number of community members participants in FGD/ total HCP in FGD; and total IDI “ IDIs were conducted with eight key informants (KIs) (n= XX) .

6. Line 126: “Theoretical framework” can fit in the Methods section and should not be a standalone section. Even if you present other frameworks you should stick to one (and I think this is what you did) and the sub-title should be called “theoretical framework” (not frameworks) .

7. Under the theme Affordability you include ability (Line 368) Ability to access FP service and contraceptive methods. The first sub-theme is the user’s ability to access SRH services of their own choice – referred to as ‘individual power’ by community participants. The participants described their limited financial situation (I would add - as one factor) constraining their ability to obtain contraception.

In my view ability or individual power is a much broader theme and the financial constraint is only one factor. Did other ability factors emerge from the interviews?

8. The acceptability component is not addressed: why? And if no material on that it should be mentioned in the study limitations.

Discretionary comments

1. Abstract - methods: please provide the total number of participants - community members (n=XX) and health care providers (n=XX)

2. Page 4 lines 56-58: “…Sexual and Reproductive Health (SRH)” and “Human Rights” : please avoid to use too many capital letters for names (even before using an acronym) unless they are organizations or recognized as needing a capital letter.

3. Discussion: you could possibly discuss the complexity of the term access, its definitions or different meanings?

Reviewer #4: This is a revised version. This research has been well done. The reviewers from the previous round made important comments, but the authors seem to me to have replied adequately. I have only a few small comments or suggestions:

1. The section titled ‘analysis’ is about data-collection and data-analysis. I suggest to split this into two sections.

2. Is it possible to reflect on the fact that selection bias can occur during the recruitment of the participants? Can this influence the results?

7. PLOS authors have the option to publish the peer review history of their article (what does this mean?). If published, this will include your full peer review and any attached files.

Reviewer #3: **Yes: **Therese Delvaux

Reviewer #4: **Yes: **Wim Peersman

---

## [Author Response · Author response to Decision Letter 1]

11 Jan 2023

Reviewer 3

1. This paper attempts to fill a gap which is the lack of use of modern contraceptive methods in KwaZulunatal, South Africa. The paper is well written and qualitative data shed some light on “access” to family planning methods from a broader perspective that include five elements according to a theoretical framework. However, these five elements needs to be well defined from the start. Unfortunately, data don’t bring much new insights in particular on the acceptability issue including the issue of secondary effects which is reported worldwide as the main reason for lack of uptake or continuation. The social factors or what is called the “social constructionism” lens used to view the data could be more prominent and data presented and discussed in light of social determinants (unmarried or single women or other vulnerable groups).

Response: 

Thank you for your review.

This study specifically focused on access to contraception in the South African public sector clinics. No previous studies have been conducted that specially focused on access to contraception using the five A’s framework. The aim of the study, and this manuscript, was to uncover the barriers and enablers to accessing contraception in this setting. The aim was not to further the definition of access or its components. 

In the previous draft of this manuscript, reviewer 2 suggested that the definitions are not necessary and lengthen the overall paper. The theoretical framework section was therefore shortened and the definitions removed. I have, however, reinserted these definitions under the theoretical frameworks section again. Please see lines 132-139

The acceptability component was explored in detail in another publication as the data fit more accordingly under the quality of care framework. Please see this publication for further reading:

Kriel, Y., Milford, C., Cordero, J. P., Suleman, F., Steyn, P. S., & Smit, J. A. (2021). Quality of care in public sector family planning services in KwaZulu-Natal, South Africa: a qualitative evaluation from community and health care provider perspectives. BMC Health Services Research, 21(1). https://doi.org/10.1186/s12913-021-07247-w

The social constructionist stance was used to analyze and review the data. Due to word length limitations and revisions requested from the previous reviewer, we are unable to fully explore the social constructionist viewpoint in this paper. However, important social structures are highlighted such as the various group categories (married, unmarried, no children, and men.). The social constructionist stance was very helpful as a lens through which to view the data, especially since there are significant social hierarchies between the groups of participants in the study. For instance, we were consciously aware of the power dynamics that can exist between healthcare providers and community members and that the healthcare provider’s view can sometimes override that of community members/clients. This power dynamic also exists between the adult and adolescent participants. 

Therefore, social constructionism provided guidance in minimizing biases in our reporting. 

2. There is some confusion or contradictions along the text with the term “access” despite the fact that you present a framework you will use in this study and you define access accordingly.

It should clearly presented from the start in the theoretical framework section after defining access, what each term (Five interrelated components of access used in this framework, namely accessibility, availability, affordability, accommodation, and acceptability) will entail, stick to that and be consistent.

Response:

Thank you, the definitions were re-inserted.

Careful attention was paid throughout this study to the definitions of the components of access. 

3. Page 2 lines 47-48 it is mentioned :” This study illustrated the importance of examining access as a holistic concept and not just the availability of services or methods.” I don’t think that access is commonly related or limited to the availability of contraceptive methods; on the contrary access refers primarily to geographical and financial access. I would remove or reformulate that sentence.

Response:

In our read of the literature, access was commonly equated with availability. The sentence was updated. 

4. Moreover in the introduction page 4 lines 96-100: you had a contradictory statement as a justification of the paper: “Only focusing on affordability and physical access contributes to an inaccurate understanding of access. Other sources have identified additional barriers, including poor quality of care (QoC), limited variety of modern contraceptive methods, fragmented integration of sexual and reproductive health (SRH) services with PHC services, ailing infrastructure, shortage of trained nurses, poor attitudes of nurses, and stigma (5, 15-19)”

Response:

This sentence is in reference to the categories used in the DHS survey to measure access. 

5. Results : Line 280 Accessibility: “Three sub-themes emerged from this theme – geographic access to current contraceptive sources, infrastructure, and preferred source of contraceptive methods.” Then later line 296 “ Infrastructure : HCP participants described the availability and accessibility of contraceptive methods at their healthcare facility related to the available infrastructure.

Response:

This sentence was updated. Please see lines 306-308.

6. Again in the Conclusion there is some contradiction and confusion regarding the meaning of access: It is stated line 616 “Overall, the participants were satisfied with the accessibility and affordability components” and later line 623 “ Men described the financial burden of accessing contraception and highlighted the importance of having an alternative source of contraception in the communities that are easily accessible.” Please clarify and reformulate

Please define the terms from the start and harmonize the text and the abstract.

Response:

The first sentence refers to the overall study findings. However, male participants added a specific view on the cost of accessing contraception. This cost is not often recognized or reported. It is, therefore, important to point out these differences. 

The sentence was slightly altered for clarity. 

7. Page 5 : the second paragraph (Lines 105-120) is long and could be quite shortened if you start the paragraph by the following : “Despite the enabling policies and restructuring, the SA public health sector has struggled to 118 make real gains in improving access to services, including FP service (20, 29, 30).

Response: 

Thank you for the suggestion, however, the preceding sentences provide the context with which to justify the last statement. 

8. Introduction: Although your study focuses on the public sector, it would be indicated to mention in the introduction (and also in the settings- eThekwini District of KwaZulu-Natal SADHS results related to what are the main sources of family planning in SA, in particular: % of public vs. private services? , and most preferred sources of contraception among young people and adults ? and most contraceptive methods used in SA and KwaZulu-Natal.

Response: 

A paragraph was inserted regarding the sources of contraception, see lines 105 - 119

9. Page 8 “CPR, “for KZN has consistently fallen below the national median, with the 2018/19 CYPR 154 being 44.4%” then later “The CYPR for eThekwini is also below the national average 159 at 44.4% for 2018/2019” please clarify for consistency. Page 8 lines 155: Study settings: Please add HIV prevalence (then the total population affected by HIV) in KwaZulu natal.

Response:

Thank you for seeing this error – the second sentence is an error and was deleted. 

Please see lines 175-179 for the HIV statistics.

10. Page 8 line 171 “Twelve of these FGDs were with community participants (n= XX) , and two with HCP (n= XX) ” I think you removed the table (duplication of data presentation) but please mention the total number of community members participants in FGD/ total HCP in FGD; and total IDI “ IDIs were conducted with eight key informants (KIs) (n= XX) .

Response:

Thank you for noticing this omission. The reviewer 2 suggested a removal of the tables that describe the participant characteristics to reduce data duplication. I have re-inserted the total numbers for each category. 

11. Line 126: “Theoretical framework” can fit in the Methods section and should not be a standalone section. Even if you present other frameworks you should stick to one (and I think this is what you did) and the sub-title should be called “theoretical framework” (not frameworks).

Response:

In the previous review, reviewer 2 requested that the theoretical framework section be separate. I have made the edits as you suggested and moved it back to the methodology section.

12. Under the theme Affordability you include ability (Line 368) Ability to access FP service and contraceptive methods. The first sub-theme is the user’s ability to access SRH services of their own choice – referred to as ‘individual power’ by community participants. The participants described their limited financial situation (I would add - as one factor) constraining their ability to obtain contraception.

In my view ability or individual power is a much broader theme and the financial constraint is only one factor. Did other ability factors emerge from the interviews?

Response:

Thank you for your comment.

I agree that ability could be regarded as a broader theme. However, specific to this data, the participants referenced their financial situation and ability to pay, as key factors that influence their access to services. 

I have edited the theme heading to clarify that in this paper I am referring to the financial/economic ability to access services. 

13. The acceptability component is not addressed: why? And if no material on that it should be mentioned in the study limitations.

Response:

Thank you for this comment.

In the overall project, the Quality of Care was also examined as a separate topic. Considering the definition of the acceptability component it was more fitting to report the acceptability data in a separate paper under the quality of care framework. This is highlighted in lines 300-301. For further reading on this, please see out other publication: (Kriel et al., 2021) 

Kriel, Y., Milford, C., Cordero, J. P., Suleman, F., Steyn, P. S., & Smit, J. A. (2021). Quality of care in public sector family planning services in KwaZulu-Natal, South Africa: a qualitative evaluation from community and health care provider perspectives. BMC Health Services Research, 21(1). https://doi.org/10.1186/s12913-021-07247-w

14. Abstract - methods: please provide the total number of participants - community members (n=XX) and health care providers (n=XX)

Response:

Thank you, the total numbers were added.

15. Page 4 lines 56-58: “…Sexual and Reproductive Health (SRH)” and “Human Rights” : please avoid to use too many capital letters for names (even before using an acronym) unless they are organizations or recognized as needing a capital letter.

Response: 

Thank you, the text was updated.

16. Discussion: you could possibly discuss the complexity of the term access, its definitions or different meanings?

Response:

The aim of this paper was not to address the complexity of access as a concept, however, I added a short paragraph in the discussion which highlights this. Please see lines 509-524.

Reviewer 4

This is a revised version. This research has been well done. The reviewers from the previous round made important comments, but the authors seem to me to have replied adequately. I have only a few small comments or suggestions:

1. The section titled ‘analysis’ is about data-collection and data-analysis. I suggest to split this into two sections. 

Response:

Thank you for the suggestion. The sections were split and the headings were changed. 

2. Is it possible to reflect on the fact that selection bias can occur during the recruitment of the participants? Can this influence the results?

Response:

Thank you for this comment. The selection bias was minimized in this study as the enrolment consisted of community members who may or may not have been contraceptive users. Random community members were recruited into this study instead of only clinic attendees. However, a small bias may well be present as only members from two communities were selected. A short paragraph was added to highlight this – please see lines 638-641.

---

## [Decision Letter · Decision Letter 2]

27 Feb 2023

PONE-D-21-18604R2Access to public sector family planning services and modern contraceptive methods in South Africa: A qualitative evaluation from community and health care provider perspectives.PLOS ONE

Dear Dr. Kriel,

Thank you for submitting your manuscript to PLOS ONE. After careful consideration, we feel that it has merit but does not fully meet PLOS ONE’s publication criteria as it currently stands. Therefore, we invite you to submit a revised version of the manuscript that addresses the points raised during the review process.

The authors have adequately responded to the comments of the reviewers. However, I have a few additional edits I would like the authors to make. Only numbers 1, 2 and 7 are required. The other comments are suggestions to improve the clarity and readability of an already well written manuscript. Lines 120-121: “On a national policy level, the SA government is dedicated to improving access to SRH services and modern contraception by committing to the goals set out by FP2020 (24).” Have these goals been carried over by FP2030? Please edit this sentence reflecting the transition from FP2020 to FP2030.Lines 129-130: “However, barriers to accessing health care services (as mentioned above) must be addressed for integration to be effective (WHO, 2008).” Please use the Vancouver style to reference this WHO document as has been done in the rest of the manuscript.Lines 175 & 706: Please edit “The couple’s year of protection rate...” to read “The couple-years of protection rate...”.Lines 192-193: For clarity, I suggest the number of FGDs (n=14) be written before the number of individuals that participated in the FGDs (n=127). Please consider this suggestion in the abstract as well.Lines 229-230: Please delete the word “study” so the sentence reads, “During the FGDs and IDIs, participants were asked to explore their views about the uptake of modern contraception.”Lines 279-280:  I would argue that 53% is not “most”. I suggest the authors consider “over half” or “Fifty-three percent” or another phrase or word.Please take a look at the references listed below and edit accordingly. I feel each of them needs a minor edit, such as a space, the insertion of the journal volume or number, or the deletion of a duplicated year. References: 18 (lines 761-762), 24 (lines 775-776), 37 (lines 808-809), 40 (lines 815-816), 43 (lines 822-823), 46 (line 831), 56 (line 855), 62 (line 872), 65 (line 879).

We look forward to receiving your revised manuscript.

Kind regards,

Funmilola M. OlaOlorun, PhD

Academic Editor

PLOS ONE

Journal Requirements:

Reviewers' comments:

Reviewer's Responses to Questions

**Comments to the Author**

1. If the authors have adequately addressed your comments raised in a previous round of review and you feel that this manuscript is now acceptable for publication, you may indicate that here to bypass the “Comments to the Author” section, enter your conflict of interest statement in the “Confidential to Editor” section, and submit your "Accept" recommendation.

Reviewer #4: (No Response)

2. Is the manuscript technically sound, and do the data support the conclusions?

Reviewer #4: Yes

3. Has the statistical analysis been performed appropriately and rigorously? 

Reviewer #4: N/A

4. Have the authors made all data underlying the findings in their manuscript fully available?

Reviewer #4: No

5. Is the manuscript presented in an intelligible fashion and written in standard English?

Reviewer #4: Yes

6. Review Comments to the Author

Reviewer #4: I still have 2 comments:

1. Lines 229 – 251 are about data collection and not data-analysis.

2. It is unclear how the participants were recruited. On lines 193 – 194 the authors wrote: “Purposive snowball sampling was used as the recruitment strategy”, while on lines 638 – 639, they wrote: “While the selection of the community and healthcare participants into this study was random a selection”.

7. PLOS authors have the option to publish the peer review history of their article (what does this mean?). If published, this will include your full peer review and any attached files.

Reviewer #4: **Yes: **Wim Peersman

---

## [Author Response · Author response to Decision Letter 2]

28 Feb 2023

Responses to reviewers: 27 Feb 2023

1. Lines 120-121: “On a national policy level, the SA government is dedicated to improving access to SRH services and modern contraception by committing to the goals set out by FP2020 (24).” Have these goals been carried over by FP2030? Please edit this sentence reflecting the transition from FP2020 to FP2030.

Response: - Sentence was edited to update to FP2030

2. Lines 129-130: “However, barriers to accessing health care services (as mentioned above) must be addressed for integration to be effective (WHO, 2008).” Please use the Vancouver style to reference this WHO document as has been done in the rest of the manuscript.

Response: Thank you for noticing this. The reference was updated to Vancouver style. 

3. Lines 175 & 706: Please edit “The couple’s year of protection rate...” to read “The couple-years of protection rate...”.

Response: Thank you for noticing this error. Update made. 

4. Lines 192-193: For clarity, I suggest the number of FGDs (n=14) be written before the number of individuals that participated in the FGDs (n=127). Please consider this suggestion in the abstract as well.

Response: Thank you, this edit was made. It does add some clarity. 

5. Lines 229-230: Please delete the word “study” so the sentence reads, “During the FGDs and IDIs, participants were asked to explore their views about the uptake of modern contraception.”

Response: Thank you, the word study was deleted. This does make the sentence more clear. 

6. Lines 279-280: I would argue that 53% is not “most”. I suggest the authors consider “over half” or “Fifty-three percent” or another phrase or word.

Response: Thank you for the suggestion, the edit makes sense. 

7. Please take a look at the references listed below and edit accordingly. I feel each of them needs a minor edit, such as a space, the insertion of the journal volume or number, or the deletion of a duplicated year. References: 18 (lines 761-762), 24 (lines 775-776), 37 (lines 808-809), 40 (lines 815-816), 43 (lines 822-823), 46 (line 831), 56 (line 855), 62 (line 872), 65 (line 879).

Response: Thank you for noticing these errors. I have updated the references and tried to correct and edit for spacing. During the format editing by the journal the spacing should also resolve.

Reviewer #4: I still have 2 comments:

1. Lines 229 – 251 are about data collection and not data-analysis.

Response: I have updated the heading to reflect data collection and analysis. 

2. It is unclear how the participants were recruited. On lines 193 – 194 the authors wrote: “Purposive snowball sampling was used as the recruitment strategy”, while on lines 638 – 639, they wrote: “While the selection of the community and healthcare participants into this study was random a selection”.

Response: Thank you for pointing that out. Participants were selected using a snowball technique, which was purposive due to the categories required for the study. The sentence in concluding paragraph was inaccurate. I have updated that sentence to clarify the potential for study bias that could have been introduced into the data due to the selection of participants. I hope this clarifies the issue.

---

## [Editor Report · Decision Letter 3]

1 Mar 2023

Access to public sector family planning services and modern contraceptive methods in South Africa: A qualitative evaluation from community and health care provider perspectives.

PONE-D-21-18604R3

Dear Dr. Kriel,

We’re pleased to inform you that your manuscript has been judged scientifically suitable for publication and will be formally accepted for publication once it meets all outstanding technical requirements.

Kind regards,

Funmilola M. OlaOlorun, PhD

Academic Editor

PLOS ONE
---

## [Editor Report · Acceptance letter]

9 Mar 2023

PONE-D-21-18604R3 

Access to public sector family planning services and modern contraceptive methods in South Africa: A qualitative evaluation from community and health care provider perspectives. 

Dear Dr. Kriel:

I'm pleased to inform you that your manuscript has been deemed suitable for publication in PLOS ONE. Congratulations! Your manuscript is now with our production department. 

Kind regards, 

on behalf of

Dr. Funmilola M. OlaOlorun 

Academic Editor

PLOS ONE